# Aging of Bioactive Glass-Based Foams: Effects on Structure, Properties, and Bioactivity

**DOI:** 10.3390/ma12091485

**Published:** 2019-05-07

**Authors:** Pier Francesco Menci, Andrea Mari, Cindy Charbonneau, Louis-Philippe Lefebvre, Luigi De Nardo

**Affiliations:** 1Department of Chemistry, Materials and Chemical Engineering “G. Natta”, Politecnico di Milano, Piazza L. Da Vici 32, 20131 Milano, Italy; pierfrancesco.menci@mail.polimi.it (P.F.M.); andrea.mari@mail.polimi.it (A.M.); 2National Research Council Canada, Boucherville, QC J4B 6Y4, Canada; cindy.charbonneau@cnrc-nrc.gc.ca (C.C.); louis-philippe.lefebvre@cnrc-nrc.gc.ca (L-P.L.)

**Keywords:** bioglass, bioactive glass, scaffolds, aging, carbonates, bioactivity

## Abstract

Bioactive glasses (BG) possess significant bone-bonding and osteogenic properties that support their use for bone defects repair in orthopaedic and dental procedures. Recent advancement enables the manufacturing of BG-based scaffolds providing structural support during bone regeneration. Despite the wide number of studies on BG and BG-based materials, little information on their aging mechanisms and shelf life is available in the literature. In this study, the evolution of chemical species on BG-based foams was investigated via accelerated tests in the presence of CO_2_ and humidity. The aging process led to the formation of carbonates (Na_2_CO_3_ and CaCO_3_) and hydrocarbonates (NaHCO_3_). The amount and composition of nucleated species evolved with time, affecting the structure, properties, and bioactivity of the scaffolds. This study provides a first structured report of aging effects on the structure and chemico-physical properties of bioactive glass-based scaffolds, offering an insight about the importance of their storage and packaging.

## 1. Introduction

The treatment of traumatic and pathological conditions that do not lead to spontaneous bone regeneration represents a major clinical challenge [1]. The use of autologous bone graft is still considered as the gold standard procedure: it promotes bone regeneration by stimulating osteoconduction, osteoinduction, and osteogenesis [1] without the risk of immune rejection. Despite these benefits, autografts show major drawbacks. The bone harvest procedure requires an extra surgery with an important drawback related to bone availability for large reconstructions, increasing the risk of postoperative complications [2,3,4]. An alternative approach relies on the use of three-dimensional scaffolds able to support the regeneration of new bone [5]. The scaffold should be designed to maintain a controlled degradation kinetics in order to reduce the inflammatory response and to provide suitable mechanical properties during the healing process [6].

Bioactive glasses and bioactive glass–ceramics have been widely studied as bone substitute materials [7]. A rapid surface reaction occurs within 24 h after implantation, leading to the formation of a biologically active hydrocarbonate apatite (HCA) layer, with a chemical composition similar to that of the inorganic portion of mineralized bone [8,9]. Although this material provides several benefits for bone reconstruction and has been used extensively in the form of granules or putty, 3D bioactive glass-based scaffolds have not been widely used yet in the clinic. Several efforts have addressed the manufacturing of such structures through technologies enabling the large-scale production of highly reproducible and reliable implants. Different comprehensive research works [10,11,12] and reviews [9,13,14] of all technological aspects relevant to the fabrication of bioactive glass scaffolds and the resulting properties have been published, elucidating the importance of selecting processes that match the biological needs during tissue regeneration [15]. A bioactive glass-ceramic scaffold has been recently developed using a process combining powder and polymer foam technologies, with the resulting scaffolds possessing structure and porosity ideal for bone reconstruction [16].

Despite the fact that there is a growing interest in bioactive glass and derived glass-ceramic structures [9] and their in vitro and in vivo properties are well documented [14], little information on their aging mechanisms is available in the literature. Glass powders reactivity with the atmosphere has been reported by Tognonvi et al. [17]. In particular, the interaction of a soda-lime-silica glass with atmospheric CO_2_ and water (formed by the condensation of air moisture on the glass surface) was reported to result in the formation of crystalline products, such as carbonates and hydrocarbonates.

In this study, the evolution of the chemical species on the surface of scaffolds aged in atmosphere containing humidity and CO_2_ was investigated, together with the impact of these species on the properties of bioactive glass-based foams. Physical and chemical characterizations were used to follow the kinetics of carbonate formation during the aging process. Morphological characterization and uniaxial compression test provided insights in the evolution of the scaffold structure and its mechanical properties with time. Finally, the impact of aging on scaffold bioactivity was assessed by immersion tests in simulated body fluid (SBF).

## 2. Materials and Methods

### 2.1. Production of Bioglass^®^-Based Glass-Ceramic Scaffold

Bioactive glass-based glass-ceramic scaffolds were produced via a foaming approach using 45S5 Bioglass^®^ powder (original composition of 45SiO_2_–24.5Na_2_O–24.5CaO–6P_2_O_5_ wt.%) [18]. Bioactive glass (Vitryxx^®^ Bioactive Glass, SCHOTT AG, Mainz, Germany) was dry-mixed with a solid polymeric binder (phenolic resin, Varcum^®^, Durez Corporation, Addison, TX, USA) and a foaming agent powder (4-methylbenzenbesulfonhydrazide, Celogen TSH, Crompton Corporation, Middlebury, CT, USA) in a ratio of 45:54.5:0.5 wt.%. The resulting mixture was molded into the desired shape and heat-treated by a three-step thermal treatment consisting of foaming, debinding, and sintering. During foaming, the binder was melted to create a suspension charged with the glass particles. The decomposition of the foaming agent created the interconnected porosity. The debinding process, conducted at 550 °C for 4 h, allowed burning out the polymeric binder. Finally, the sintering, performed at 1035 °C for 1 h, consolidated the bioactive glass and provided mechanical properties to the final scaffold [19].

### 2.2. Preparation and Aging of Bioglass^®^ Foam Samples

Scaffolds with an average porosity of 63% and pore size distribution within the 50–750 µm range were machined into cylinders (9 mm in diameter) and manually cut with a circular diamond saw into discs of 6 mm in thickness. The porosity  p was evaluated according to Equation (1):(1)$$p=\left(1-\frac{\rho_{foam}}{\rho_{solid}}\right)\;\times \;100$$
where $\rho_{solid}$ = 2.7 g cm^−3^ is the theoretical density of 45S5 Bioglass^®^ [20].

To assess the effect of aging on foam properties, the samples were subjected to accelerated aging treatments in an incubator (Forma Series II, Thermo Electron Corporation, Waltham, MA, USA) at 37 °C, 90% relative humidity (RH), and 5% CO_2_ for up to 8 weeks. Ten samples for each aging time point were used. The control (pristine) specimens were sealed and maintained under vacuum to be used as reference samples, while the remaining ones were placed in open glass bottles and put in the incubator for aging. After aging, the samples were dried in an oven (ISOTEMP 550D, Fisher Scientific, Hampton, NH, USA) at 100 °C for 24 h to remove residual absorbed water.

### 2.3. Preparation of Simulated Body Fluid for Bioactivity Evaluation

Simulated Body Fluid (SBF) was prepared in accordance to ISO standard 23317:2014 [21]. A polypropylene beaker was filled with 700 mL of deionized water (DI H_2_O, 18 MΩ cm, Milli-Q system, Millipore Merck, Molsheim, France). The beaker was placed in an oven, until the water temperature reached 37 °C and then shifted on a magnetic stirrer (T = 37 °C). The salts (8.035 g L^−1^ NaCl, 0.355 g L^−1^ NaHCO_3_, 0.225 g L^−1^ KCl, 0.231 g L^−1^ K_2_HPO_4_ 3H_2_O, 0.311 g L^−1^ MgCl_2_ 6H_2_O, 39 mL HCl (1 M), 0.292 g L^−1^ CaCl_2_, 0.072 g L^−1^ Na_2_SO_4_, and 6.118 g L^−1^ Tris, all from Sigma-Aldrich, Milan, Italy), were slowly added to the DI H_2_O. Drops of HCl (1 M) were added until pH 7.4 was reached. DI H_2_O heated at 37 °C was then added to the solution to reach a final volume of 1 L. Finally, the solution was cooled down to room temperature, filtered to eliminate possible impurities, and stored in a closed bottle at 4 °C.

### 2.4. Characterization of Bioglass^®^ Foam Samples

#### 2.4.1. Chemical and Physical Characterization

The evolution of the foam was evaluated using mass uptake, carbon content using elemental analysis, X-Ray diffraction spectroscopy (XRD), mid-infrared reflectance (MIR), and pH analysis (during immersion in water and SBF). Mass uptake (MU) was evaluated after aging and drying to estimate the formation of new species on the sample surface using Equation (2): (2)$$\mathrm{MU}\left(t\right)=\left(\frac{w_{d}\left(t\right)-w_{0}}{w_{0}}\right)\;\times \;100$$
where $w_{0}$ is the initial weight, and $w_{d}\left(t\right)$ is the weight after drying at each time point *t*.

A carbon determinator (LECO CS600, St Joseph, MI, USA) was used to evaluate the carbon uptake after aging and drying. The results were obtained by performing the same measurement three times on 0.5 g of material for each sample.

The evolution of the crystalline phases was followed using XRD (D8 Discover, Bruker, Billerica, MA, USA) with Cu kα radiation. The patterns were recorded on the 20°–90° 2θ range, using 0.02° step size and analyzed with the software DIFFRAC.EVA V2.0 (Bruker, Billerica, MA, USA).

Further information on the foam surface species was collected by MIR in the 750–1900 cm^−1^ range. The MIR spectra were measured using a bidirectional laser-based reflectance setup with a tunable external cavity quantum cascade laser (Block Engineering, LaserTune, Marlborough, MA, USA, wavelength of 5.2 μm to 13.4 μm) used and operated at a pulse rate of 10 kHz and a pulse duration of 100 ns. The MIR laser beam was focused on the sample with a spot area of 400 × 500 µm^2^ using a gold parabolic mirror at normal incidence. The working distance was 10 cm. The reflected light was collected and focused on a 1 × 1 mm^2^ HgCdTe (MCT) detector (Infrared Associate, FTIR-16, Stuart, FL, USA). The detector was operated at 77 K using liquid nitrogen. The spectra were formed by continuously tuning the emission wavelength and by recording each pulse amplitude value at different wavelengths. The raw spectra were smoothed using a moving average filter and normalized using the reflectance spectrum of a KBr pellet. For each sample, 36 spectra were taken in a 6 × 6 square pattern with a spacing of 0.75 mm.

pH measurements were performed using a pH meter (Seven Excellence, Mettler Toledo, Columbus, OH, USA). For each time point, a foam sample was crushed and immersed in DI H_2_O at a concentration of 1.5 g L^−1^. A magnetic stirrer was used during pH readings with a speed of 100 rpm to obtain a homogeneous dissolution.

#### 2.4.2. Morphological Characterization

The morphology evolution of the scaffolds was assessed using scanning electron microscopy (SEM), and micro-computed tomography (µCT). SEM (S-4700, Hitachi, Tokyo, Japan) was used in high-vacuum mode at the operating voltage of 2 kV. The samples were coated with a thin layer of platinum using a sputter coater (K575X, Quorum Technologies, Lewes, DE, USA).

The foam structure was also analyzed using µCT (X-Tek, HMXST 225, Nikon Metrology, Tring, UK). The µCT scanner was equipped with a Perkin-Elmer 1621 AN amorphous silicon flat panel (409.6 ×409.6 mm^2^) coupled to a CsI scintillator. The X-Ray source was operated at 104 kV and 36 μA, with an integration time of 1000 ms, 3142 projections over 360°, and 1 frame per projection. The pore size distribution was evaluated by image analysis using a Clemex Vision PE software (V 8.0, Clemex Technologies, Longueuil, QC, Canada). The analysis was conducted on 2D sections extracted from µCT reconstructions, using nine slices in the vertical direction and three in the horizontal one for each sample. The margins of the slices were not taken into account during the analysis.

#### 2.4.3. Mechanical Characterization

The mechanical properties were evaluated with uniaxial compression tests using an MTS 10 kN compression testing machine (Instron, Norwood, MA, USA) with a crosshead speed of 2.5 mm min^−1^. The compressive strength was defined as the maximum on the stress–strain curves. Six samples for each aging time point were characterized.

#### 2.4.4. Bioactivity

Immersion tests in SBF were performed on scaffolds to investigate the effect of aging on their bioactivity. Cylindrical specimens were manually divided in quarters of disc using a saw. After cutting, powder debris and edges contaminated by the saw were removed with a file. The bioactivity investigation was conducted using 36 samples, divided into 3 groups of 12 samples each and aged for different durations: 0, 2, and 4 weeks. Each sample was immersed in SBF at a concentration of 1.5 g L^−1^ according to ISO/FDIS 23317. The samples were placed into polypropylene bottles with screw top and incubated at 37 °C up to 7 days. An oscillating platform was used to keep the SBF in agitation. After immersion, the samples were washed with DI H_2_O and dried in the ISOTEMP 550D oven (Fisher Scientific Co., Fair Lawn, NJ, USA) for 24 h at 100 °C.

The weight reduction $w_{r}$ was evaluated after immersion in SBF for each sample using the Equation (3): (3)$$w_{r}\left(t\right)=\left(\frac{w_{0}-w_{i}\left(t\right)}{w_{0}}\right)\;\times \;100$$
where $w_{0}$ is the weight of unsoaked samples, and $w_{i}\left(t\right)$ the weight after drying at each time point *t*.

A pH meter (HACH HQ11d, Loveland, CO, USA) was used to evaluate the pH at different time points. 

The surface of the foam after immersion in SBF was analysed using an SEM (EVO 50, Carl Zeiss, Oberkochen, Germany) coupled with an energy dispersion X-Ray spectroscope (EDS, Oxford INCA 200, Oxford Instruments, Abingdon, UK) and XRD (Empyrean, Malvern Panalytical, Malvern, UK) with Cu kα radiation in the 2θ range of 20°–65° using 0.02° step.

## 3. Results

### 3.1. Chemical and Physical Characterization

Mass and carbon uptakes of foams aged at 37 °C, 90% RH, and 5% CO_2_ are presented in Figure 1. A monotonic increase of mass with aging was observed; a mass uptake up to 12.2% ± 1.0% after 8 weeks of aging was measured (Figure 1a). The carbon content also increased up to 4.1% ± 0.2% after 8 weeks of aging (Figure 1b).

The XRD patterns of aged scaffolds are reported in Figure 2a. The unaged (pristine) sample showed the typical composition of a glass-ceramic material, as reported in previous studies [7]. Accelerated aging led to the nucleation of calcium carbonate (CaCO_3_) after 1 week, being still present after 8 weeks. Sodium bicarbonate (NaHCO_3_) was detected after 1 and 2 weeks of aging, while sodium carbonate (Na_2_CO_3_) was observed after 4 weeks. These results were confirmed by MIR (Figure 2b); the peaks around 840 cm^−1^ on the MIR spectrum corroborated the presence of NaHCO_3_, while the peaks around 880 cm^−1^ corresponded to CaCO_3_ and/or Na_2_CO_3_.

The variations of the pH of DI H_2_O solutions with foams aged for different period of time are shown in Figure 3. The pH values of DI H_2_O solutions with foams aged 1 and 2 weeks were lower (pH = 9.8) than the pH of the solutions with unaged samples (pH = 10.5) and with samples aged 4 to 8 weeks (pH = 10.9).

### 3.2. Morphological Characterization

Figure 4 presents surface micrographs of the foams before and after aging, as observed by SEM at different magnifications. The unaged sample (Figure 4a,b) showed a pristine surface. Carbonates nucleated on the surface and filled the pores of the foams as the aging time increased. A leaf-like structure was observed after 1 week of aging (Figure 4d), growing into a needle-like shape after 8 weeks aging (Figure 4f).

The analysis of the foam structure on 2D images extracted from μCT reconstructions (Figure 5a,c,e) suggests an effect of aging on pore size distribution. The cumulative pore size distribution curves moved toward a higher pore size during the first weeks (Figure 5d) and shifted toward a lower pore size after 8 weeks of aging (Figure 5f). 

### 3.3. Mechanical Characterization

Figure 6a shows the typical stress–strain curves of a pristine and an aged foam (8 weeks). A brittle behavior was observed, where stress linearly increased until a critical value was reached. 

The effect of aging on the compression strength is presented in Figure 6b. The compression strength increased from 7.6 ± 1.2 MPa for the unaged foam up to 11.9 ± 2.9 MPa after 8 weeks of aging.

### 3.4. Bioactivity Evaluation

Figure 7a shows the dissolution kinetics of the scaffolds aged in SBF. The weight variation did not follow a clear trend during the first 3 days of immersion. However, for longer periods of immersion, the specimens showed a reduction of their weight of 17% after 7 days for the samples aged 4 weeks. The pH of SBF solutions increased with the aging time (Figure 7b), starting at 7.4 for SBF alone. The pH of the unaged sample reached a value of 7.9 ± 0.1 after 7 days of immersion, while the pH of the foams aged 2 and 4 weeks reached 7.8 ± 0.0 and 7.9 ± 0.0, respectively. 

Figure 8 reports the microstructural evolution the surface of bioactive glass foams after immersion in SBF for 7 days. Calcium phosphate (CaP) bulb-like structures were observed on the foam surface after immersion. Samples aged 4 weeks showed the highest apparent concentration of CaP, compared to samples aged 2 weeks and to unaged samples (Figure 8).

The formation of CaP on the surface of the foam was confirmed by EDS (Figure 9), showing the phosphorous atomic percentage increased with aging after 7 days of immersion in SBF. XRD showed a reduction of peak intensity for Na_2_Ca_2_Si_3_O_9_ [JCPDS #98-006-0502] and CaCO_3_ [JCPDS #98-002-0179] after aging (Figure 10), suggesting a decrease of crystallinity of the foam with increasing immersion time in SBF. Also, the characteristic peaks of hydroxyapatite carbonate (HAC) [JCPDS #98-015-0310] were detectable only for samples aged 4 weeks after 7 days of immersion in SBF (Figure 10). For all the other samples, it was not possible to identify any peak related to HA, despite the fact that calcium phosphates were detected by means of SEM and EDS (data not shown).

## 4. Discussion

### 4.1. Nucleation of Carbonates as a Consequence of the Aging Process

The bioactive glass-based scaffolds aged in contact with humidity and CO_2_ formed carbonate species. The mass uptake and carbon content increases were associated with the formation of carbonates on the surface of the scaffolds (Figure 11). This result was confirmed by XRD and MIR analyses. CaCO_3_ was detected throughout the aging process, while conversion of NaHCO_3_ into Na_2_CO_3_ was observed after 2 weeks of aging (Figure 2a,b).

An increase of carbonate concentration was observed as the aging time increased. The presence of crystals on the bioactive glass-based scaffold surface was already observed by C. Charbonneau et al. [22]. Carbonate concentration increased with aging, resulting in lower porosity. The shift toward higher pore size distributions (Figure 5d) during the first weeks of aging can be related to the filling of the finer pores by carbonates. After 8 weeks of aging, the curves shifted toward smaller pores, as larger pores were partially filled by carbonates (Figure 5f).

The reduction of porosity with aging affected the mechanical strength of the foams. The porosity of aged samples was evaluated using Equation (1), assuming the material had the same theoretical density of Bioglass (2.7 g cm^−3^). This approach does not take into consideration the density of the other species, i.e., CaCO_3_ (2.71 g cm^−3^), Na_2_CO_3_ (2.54 g cm^−3^), and NaHCO_3_ (2.20 g cm^−3^) [23]. This approximation was used since the relative concentration of the different species can not be evaluated with precision. This simplification should not, however, lead to wrong interpretations, as the amount of carbonate was low for short aging times. For longer aging times, there was essentially no NaHCO_3_ and comparable amounts of CaCO_3_ and Na_2_CO_3_, both having a theoretical density near the density of Bioglass. The density of the composite materials should thus be similar to the density of Bioglass. The general dependence of compressive strength on porosity is qualitatively reported in Figure 12. The compression strength increased linearly with the mass uptake of the foams, from 4.6 MPa to 16 MPa for 74% and 51% porous scaffolds, respectively (Figure 12).

Compressive strength is used as a preliminary yet powerful mechanical characterization technique, since it offers an easy way to compare the mechanical properties of the obtained materials with those of cortical and trabecular bone and structures previously published [24]. In all cases, both pristine and aged scaffolds showed mechanical compressive strength comparable to the values reported for human trabecular bone (2–12 MPa) [24].

### 4.2. Effects of the Aging Process on Bioactivity

The increase of carbonates content with aging time can potentially affect its dissolution and biocompatibility. Dissolution tests in water showed a pH decrease for short aging times, while an opposite behavior was observed for longer aging (4 and 8 weeks). As observed by Cerruti et al. [25], the fast pH increase when bioglass is immersed in DI H_2_O is related to the exchange of Na^+^/H^+^ ions and to the absence of a buffer in the solution. Differences in pH behavior for aged specimens can be explained by the different quantity and nature of the carbonates formed on the surface of the foams, as supported by XRD and MIR analysis. The pH and solubility values of carbonate species are reported in Table 1. The lower pH values associated with the foams aged 1 and 2 weeks can be explained by the presence of NaHCO_3_, which has lower pH and solubility values than Na_2_CO_3_. On the contrary, higher pH values reached with samples aged 4 and 8 weeks can be explained considering a substitution of NaHCO_3_ with Na_2_CO_3_ (Figure 2a). Since the solubility of CaCO_3_ is much lower than that of the other species, its contribution on the final pH can be neglected.

The bioactive glass-based scaffold bioactivity was assessed by immersion in SBF. The weight reduction evaluated for the scaffolds after immersion was related to both their partial degradation and carbonates dissolution: a greater weight reduction was observed as the aging time increased. In particular, the foams aged 2 and 4 weeks were characterized by the presence of NaHCO_3_ and Na_2_CO_3_, respectively, whose high solubility (Table 1) was responsible for the quick weight reduction immediately after immersion in SBF (Figure 7a). CaCO_3_ presence did not significantly contribute to weight reduction because of its low solubility. 

The degradation mechanism of bioactive glass-based scaffolds in SBF has been previously described by Boccaccini et al. [26]. According to the proposed model, scaffold degradation takes place at the interface between the glass matrix and the combeite crystallites (Na_2_Ca_2_Si_3_O_9_). This phenomenon preferentially occurs at the subgrain boundaries of the crystalline phases, leading to the degradation of non-sintered crystalline structures. Furthermore, ions exchange contributes to punctual defects increase, distorting the periodic structure of the crystalline phase and leading to the amorphization. Cations exchange, in particular Na^+^ and Ca^2+^, from glass and carbonates with protons from the solution resulted in a fast increase of pH (Figure 3). In the present case, the pH increase was slowed down by the buffering capacity of SBF (Figure 7b). The trends observed for the different aging times were in accordance with the ones observed in water, with a lower pH value, after 2 weeks of aging with respect to the unaged sample. A higher pH value obtained for the foams aged 4 weeks was confirmed as well. The pH increase was buffered by the SBF, resulting in pH values that were comparable to those obtained with the aged materials. This buffering is important for in vivo applications, as high pH values can be toxic for cells. It is important to specify that physiological fluids also have buffering capabilities and will temper the pH of surrounding tissues.

Phosphorous concentration provides an indication of HA concentration on the surface of the scaffolds. Figure 9 shows a phosphorous reduction for the first 3 h of immersion in SBF. This reduction seemed to be less important as the foam aging increased. A different trend was observed after 7 days, indicating that HA formation was promoted by aging. The initial low bioactivity observed for the aged samples can be related to the fast NaCO_3_ and NaHCO_3_ dissolution, which hindered HA nucleation during the first few hours of immersion. However, carbonates decomposition increased the concentration of ions in solution, contributing to HA deposition at longer immersion times.

The bioactive glass scaffolds exhibited diffraction peaks attributed to Na_2_Ca_2_Si_3_O_9_, CaCO_3_, Na_2_CO_3_, and NaHCO_3_ phases (Figure 2a). The peaks related to Na_2_CO_3_ and NaHCO_3_ were not detectable after 3 h of immersion in SFB. In contrast, the CaCO_3_ phase was still present after 7 days (i.e., 168 h) of immersion because of its low solubility (Figure 10). Crystallinity reduction of sintered bioactive glass with increasing immersion time was previously observed by Boccaccini et al. [26] in a study showing the complete disappearance of the combeite crystalline phase after immersion in SBF, substituted by an amorphous phase. The fact that HA could not be detected by XRD in most of the samples (apart from the samples aged 4 weeks after 7 days of immersion in SBF) could be related to either the thickness of the layer that was probably too low to be detected by XRD or to its amorphous structure, resulting in negligible diffraction signals.

## 5. Conclusions

This study presents the impact of aging on the structure and properties of bioactive glass-based scaffolds. Aging of bioactive glass-ceramic scaffolds leads to the formation of carbonates (Na_2_CO_3_ and CaCO_3_) and hydrocarbonates (NaHCO_3_) in the presence of CO_2_ and humidity. The carbon measured in the aged specimens was strictly related to the formation of carbonates on the surface of the scaffolds. Moreover, the presence of carbonates increased the scaffolds’ mechanical compressive strength, reduced the pH after dissolution in the presence of NaHCO_3_, and had an influence on scaffolds’ bioactivity. The results indicate that the kinetics of formation of carbonate was slow enough to prevent the formation of a large amount of carbonates if the exposure was short at room temperature.

The aging process could, nevertheless, be minimized by adequate packaging, thus preventing the reaction of glass with humidity and CO_2_. However, if no care is taken to avoid the exposure of the bioactive glass to air, aging may occur, and this may impact the properties of bioactive glass, such as its chemical composition. Aging could also have an effect on the pH after implantation or if the glass is used in suspension (e.g., putty, tooth paste, cosmetics).

Further investigations could be performed to assess the overall effects on the in vitro and in vivo behavior of aged bioactive glass and determine the effects of nucleated species.

## Figures and Tables

**Figure 1 materials-12-01485-f001:**
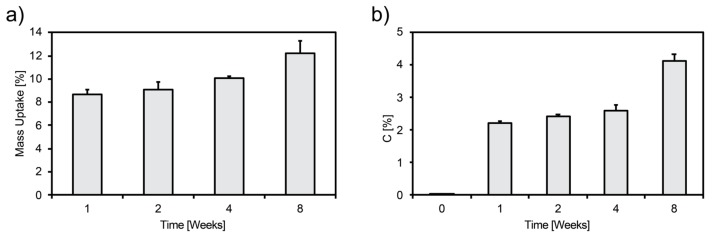
(**a**) Mass and (**b**) carbon uptake vs. aging time.

**Figure 2 materials-12-01485-f002:**
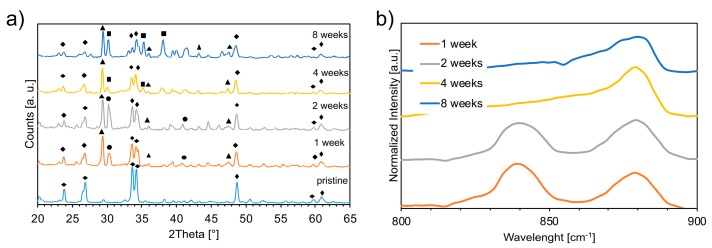
(**a**) XRD patterns of unaged foam and aged foams at different time points: 1, 2, 4, and 8 weeks. (■) = Na_2_CO_3_ (PDS 37-0451 and PDS 19-1130); (●) = NaHCO_3_ (PDS 15-0700); (▲) = CaCO_3_ (PDS 01-837, PDS 83-0577, PDS 85-1108); (◆) = Na_2_Ca_2_Si_3_O_9_ (PDS 75-1686 Combeite). (**b**) Normalized curves obtained by MIR spectroscopy on the same aged foams. The peak at 840 cm^−1^ corresponds to HCO_3_^−^, while the one at 880 cm^−1^ corresponds to ${\mathrm{CO}}_{3}^{2-}$. The normalized intensity represents the ratio between the sample reflected intensity and KBr reflected intensity. KBr powder, slightly pressed in a holder, was used as a reference [color figures online].

**Figure 3 materials-12-01485-f003:**
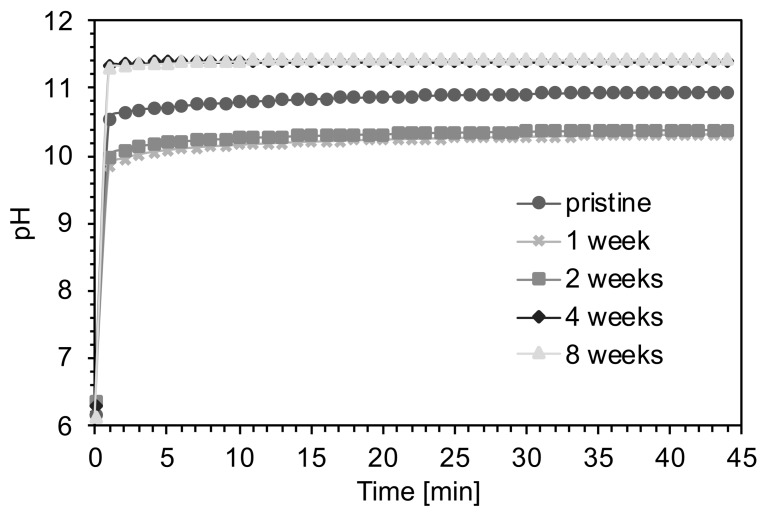
pH measurements in deionized (DI) H_2_O with foams aged at different time points (pristine, 1, 2, 4, and 8 weeks).

**Figure 4 materials-12-01485-f004:**
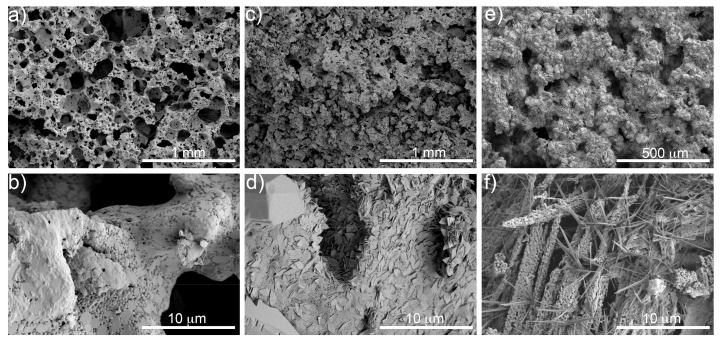
SEM micrographs at different magnifications of the unaged foam (**a**,**b**) and of foams aged 1 (**c**,**d**) and 8 weeks (**e**,**f**).

**Figure 5 materials-12-01485-f005:**
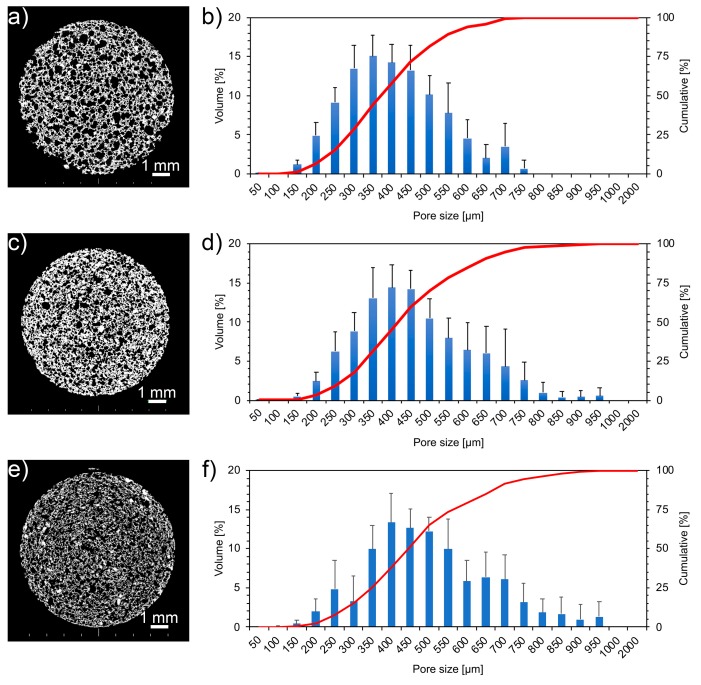
Computed micro-tomography and pore size distribution evaluated through Clemex image analysis on 2D sections of as-produced (pristine) foam (**a**,**b**), foam aged 1 (**c**,**d**), and 8 weeks (**e**,**f**).

**Figure 6 materials-12-01485-f006:**
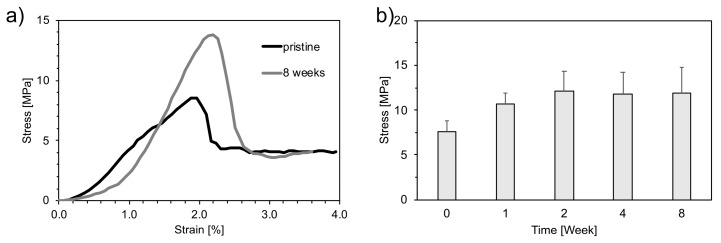
Mechanical tests: (**a**) representative stress–strain compressive behavior of unaged (pristine) and foams aged 8 weeks; (**b**) Stress at break of pristine and aged foams.

**Figure 7 materials-12-01485-f007:**
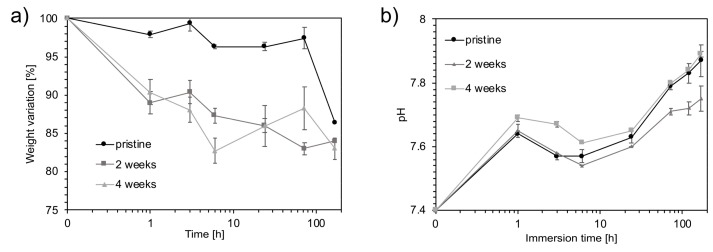
(**a**) Weight variations evaluated after 1 h, 3 h, 6 h, 24 h, 3 d, and 7 d of immersion in simulated body fluid (SBF) at each aging time point: 0, 2, and 4 weeks. (**b**) pH evaluations performed after 1 h, 3 h, 6 h, 24 h, 3 d, 5 d, and 7 d of immersion in SBF at each aging time point: 0, 2, and 4 weeks.

**Figure 8 materials-12-01485-f008:**
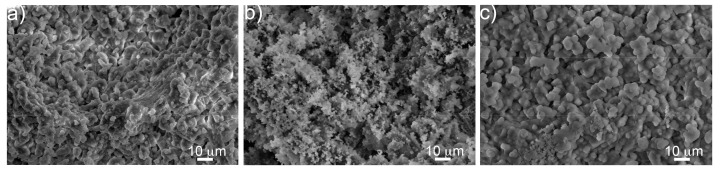
SEM micrographs of the unaged sample (**a**) and foams aged 2 (**b**) and 4 (**c**) weeks after 7 d of immersion in SBF.

**Figure 9 materials-12-01485-f009:**
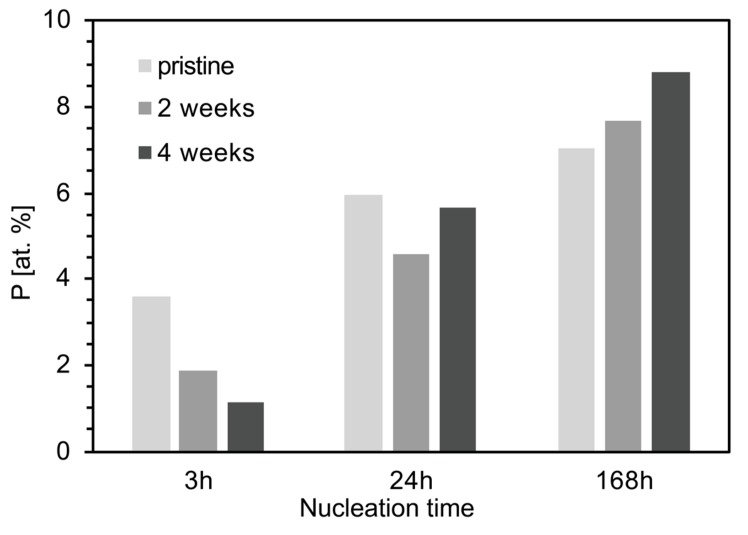
Atomic percentage of phosphorous (measured by EDS) on the surface of foams differently aged after 3 h, 24 h, and 7 days of immersion in SBF.

**Figure 10 materials-12-01485-f010:**
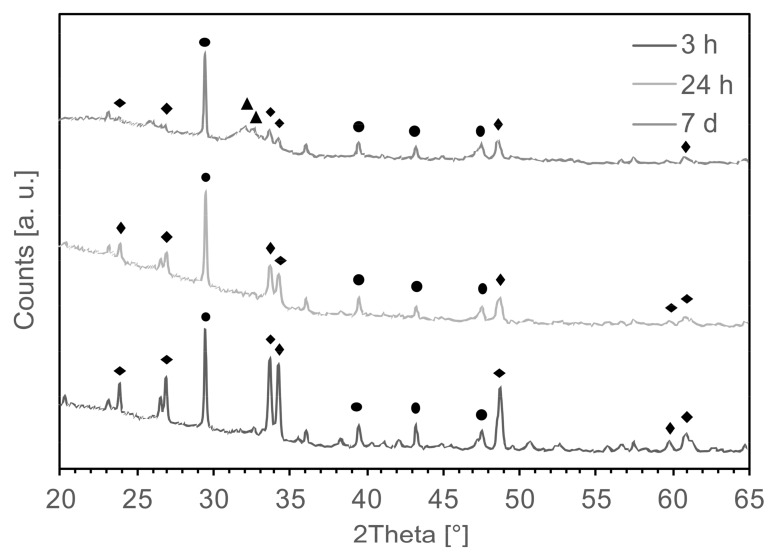
XRD analyses performed after 3 h, 24 h, and 7 d of immersion in SBF on foams aged 4 weeks. (◆) = Na_2_Ca_2_Si_3_O_9_ [JCPDS #98-006-0502]; (●) = CaCO_3_ [JCPDS #98-002-0179]; (▲) = Hydroxyapatite carbonate (HAC) [JCPDS #98-015-0310].

**Figure 11 materials-12-01485-f011:**
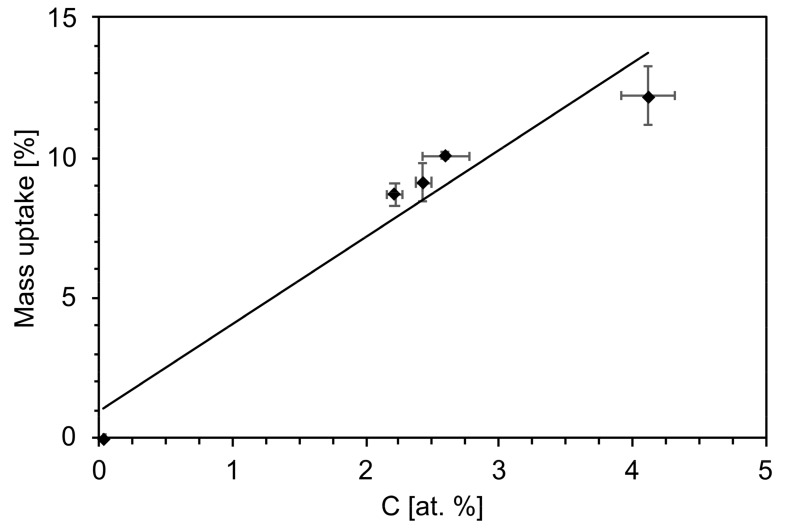
Relation between the foam mass uptake and the carbon content by increasing the aging time.

**Figure 12 materials-12-01485-f012:**
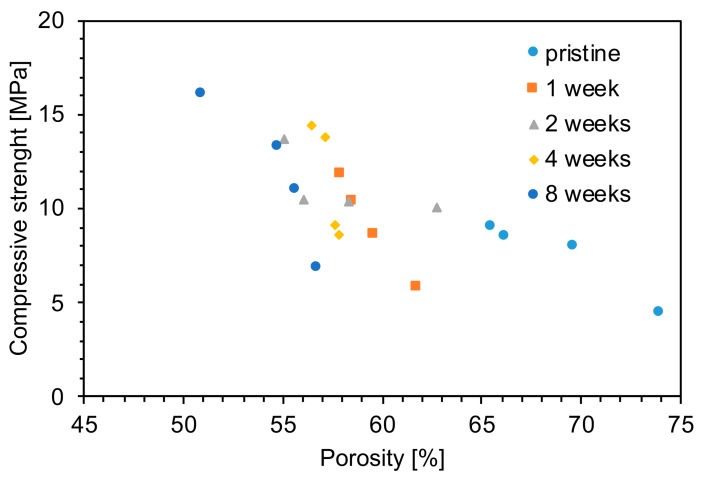
Compression strength as a function of porosity for foams aged at different time-points.

**Table 1 materials-12-01485-t001:** Solubility and pH values of Na_2_CO_3_, CaCO_3_, and NaHCO_3_ at 50 g L^−1^ concentration at 25 °C [23].

Carbonate	pH (50 g L^−1^ at 25 °C)	Solubility in Water (g L^−1^)
Na_2_CO_3_	11.5	220
CaCO_3_	9.5	0.013
NaHCO_3_	8.6	96

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
