# Peer review of "Aging of Bioactive Glass-Based Foams: Effects on Structure, Properties, and Bioactivity"

_materials, 2019, doi:10.3390/ma12091485_

Reviewer 1 Report

My comments are in the attached file.

Author Response

Please find the detailed response to the reviewer’s comments in the attached file.

Reviewer 2 Report

1) The manuscript sequence is different in journal materials. For Materials Journal - Materials and methods should come after discussion and before the conclusions. And you must separate the results from the discussion.

2) manufacturer - When mentioning materials or devices: you don't mention the manufacturer, city and the country. It would be better if you make the same mentions each time you refer to a commercial product.

Author Response

(The authors gave the same response as above.)

Reviewer 3 Report

I appreciate the authors' efforts in conducting an interesting research and reporting the results in this manuscript. The presented research is timely and possesses good scientific merit. The following are some comments that I respectfully suggest to be addressed before publication:

- The English writing of the manuscript needs to be edited, preferably by a professional editor. The narratives become hard to understand at some points and grammatical errors are quire abundant.

- Line 44: is this meant to be "bone mineral"? If yes, please revise. It would be better to provide a brief description of the so-called bone mineral's chemical nature (e.g. referring to degree of crystallinity, composition, etc.).

- A more comprehensive literature review could be done in the introduction. Related previous art has not been adequately covered. Since this is not a review paper, the authors are not expected to review the entire literature. However, providing some more insight into the papers that have been used as the basis of this research will improve the flow of the manuscript by providing a brief scientific background before the methodology. For example, major methods of fabricating bioactive glass-based scaffolds could be provided in the introduction and the reason why a specific method has been selected in this research could be described in the methodology.

- Section 2.4.3:  How is compressive strength relate to other mechanical properties of the substance? For example, how does it relate to flexural strength or fatigue life?

- Figure 4: If possible, please identify/label different phases/substances in the figure. For example, the large crystals in figure 4.b. or needle-shaped crystals in figure 4.f.

Author Response

(The authors gave the same response as above.)
